# Short-Term Forecasting of Power Production in a Large-Scale Photovoltaic Plant Based on LSTM

**Mingming Gao [1], Jianjing Li [1,*], Feng Hong [1] and Dongteng Long [2]**

[1] The State Key Laboratory of Alternate Electric Power System with Renewable Energy Sources, School of Control and Computer Engineering, North China Electric Power University, Beijing 102206, China

[2] China Academy of Aerospace Standardization and Product Assurance, Beijing 100071, China

\* Correspondence: lijianjing@ncepu.edu.cn or lijianjing95@163.com; Tel.: +86-010-6177-2568

**Abstract:** Photovoltaic (PV) power is attracting more and more concerns. Power output prediction, as a necessary technical requirement of PV plants, closely relates to the rationality of power grid dispatch. If the accuracy of power prediction in PV plants can be further enhanced by forecasting, stability of power grid will be improved. Therefore, a 1-h-ahead power output forecasting based on long-short-term memory (LSTM) networks is proposed. The forecasting output of the model is based on the time series of 1-h-ahead numerical weather prediction to reveal the spatio-temporal characteristic. The comprehensive meteorological conditions, including different types of season and weather conditions, were considered in the model, and parameters of LSTM models were investigated simultaneously. Analysis of prediction result reveals that the proposed model leads to a superior prediction performance compared with traditional PV output power predictions. The accuracy of output power prediction is enhanced by 3.46–13.46%.

**Keywords:** photovoltaic plant; power forecasting; recurrent neural networks; LSTM

---

## 1. Introduction

In the past decade, the renewable energy market is growing rapidly to reduce energy costs, greenhouse gas emissions, and the consumption of fossil fuels [1]. Solar energy is considered to be one of the most promising energy alternatives due to the sustainability and availability [2]. Photovoltaic (PV) systems are the most commonly applied in solar energy utilization [3]. As shown in Figure 1, the new installed capacity in 2017 in China was 53.06 GW, which was about 25 times of the installed capacity in 2011, and accounts for half of the global newly installed capacity. In China, the domestic PV power market is undergoing a profound change with the shift to an independent power sources.

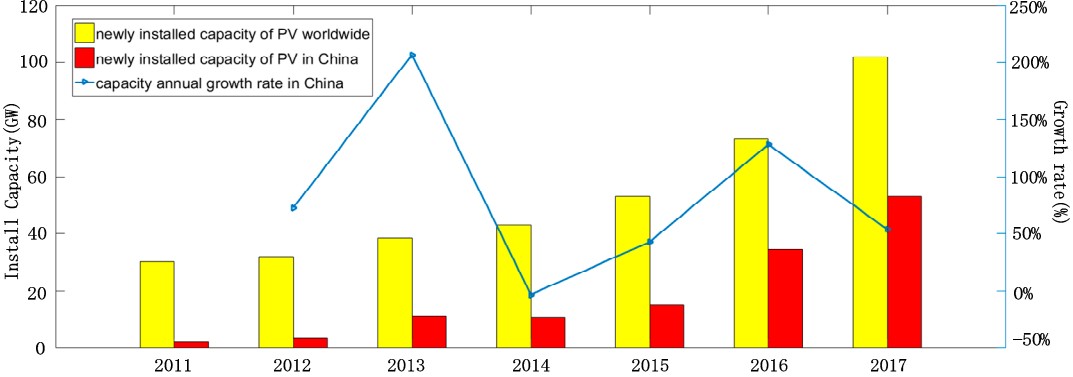

**Figure 1.** Annual newly installed capacity of PV (2011–2017).

A larger proportion of PV power in energy source structure is challenging the grid stability because of the intermittent characteristics of PV production [4]. Such unstable power production would cause control and operation problems in grids, which would lead to sudden surpluses and frequent drops in the course of grid running [5]. In order to dispatch the load reasonably and make a further dispatch plan, power output prediction for PV plants was put forward and rapidly developed in recent years [6].

With respect to time horizon of prediction, the power production forecasting of a PV plant can be grouped into two categories [7]. Long-term forecasting (from days to a month) of PV power was applied to make decisions pertaining to the installation of new PV plants and storage systems, as well as develop suitable demand response strategies [8,9]. Short-term forecasting (from minutes to 1 day ahead) was widely used in optimizing the real-time power dispatching, electricity market transaction, and maintenance plan [10]. In the view of optimizing the dispatch plan and flattening the fluctuation of PV powers, improving the precise short-term forecasting is much more significant.

It is known that the performance of a given PV plant is mainly influenced by weather conditions, especially the solar irradiance and temperature, which are intensely variable. Obviously, power output prediction of PV plants essentially based on numerical weather predicting (NWP) [11]. According to the investigating result, the accuracy of 1-h-ahead numerical weather prediction can reach 95%. However, as the prediction time horizon increases, the predict accuracy drops sharply. So the meteorological information from NWP could be accepted as the input of 1-h-ahead power forecasting.

In order to enhance predict performance, research is needed to optimize the prediction method in both input data and predict algorithm. Instantaneous monitoring data—for example, solar irradiance and solar cell or air temperature—are applied in a predictive model, real-time prediction is accurate, but the real-time data are not easily accessible [12–14]. On the other hand, based on hourly solar irradiation and air temperature data provided by weather bureau, other researchers focus on improving the prediction model. A prediction method was proposed to forecast power output of PV systems based upon the weather forecasting data and support vector machines (SVM), with the mean relative error (MRE) of 8.64% [15]. A 1-h-ahead power output forecasting of a PV system was presented using a combination of wavelet transform (WT) and radial basis function neural networks (RBFNN) by incorporating the interactions of PV system with solar radiation and temperature data [16]. The mean absolute percentage error (MAPE) values obtained from the model for sun days and cloudy days can reach 2.38% and 4.08%, respectively. However, the performance for rainy days is unsatisfactory. Another technique in these forecasting models is a specific soft-computing technique known as artificial neural networks (ANNs). Power outputs of a PV plant with forecasting horizons of 1- and 2-h-ahead were predicted with several forecasting models, and in [17] models based on ANNs optimized with genetic algorithm (GA) achieved better results. The application of artificial neural networks with a single hidden layer, which the periodicity and timing of data were not taken into account, did not greatly improve the accuracy of the prediction. Some large-scale PV systems may have various types of PV arrays installed over a wide area with different tile and azimuth angles. In addition, significant diversity of photo-electric conversion efficiency may exist among different power converters. All of these can result in potentially serious errors while modeling the specific PV power forecasting system.

An appropriate strategy to overcome the aforementioned problems could be considering the historical power output generated within 3 days. Unfortunately, only a few studies using both meteorological information of next hour from NWP and historical power output have been reported in the literature so far. Furthermore, analyzing the characteristic of historical data of PV plants, the time series of PV power generation are typical of periodicity and seasonality, the advantages of two aspects above, intelligent algorithm and time series analysis, should be combined. An intelligent algorithm which combines current information and historical effect is most compatible for PV plant power output prediction. Recurrent neural networks (RNN) algorithm is one of methods that can accommodate dependencies between consecutive time steps [18,19]. The RNN architecture structures were used for modeling time series in which neurons were fully connected with cycles feeding the activation

functions from previous time steps as inputs, and the relevance between historical data and current status could be described with this structure. However, on the other hand, a "vanishing or exploding gradients problem" coming into with this connecting structures [20].

In order to overcome the problem, recently, a new architecture for RNN has been designed called the long short-term memory (LSTM), and a novel solution which introduced 'memory blocks' was proposed to solve the problem [21]. Nowadays, LSTM algorithm has been applied to various fields, including human behavior predicting, short-term residential load forecasting and renewable energy. Aiming at early detection of the risks related to mild cognitive impairment and frailty and providing meaningful interventions that prevent these risks, [22] have created a deep learning architecture based on LSTM to predict the user's next actions and to identify anomalous user behaviors. It was also applied in residential load forecasting. The LSTM-based framework was proposed to address the short-term load forecasting problem including high volatility and uncertainty for individual residential households [23]. The proposed LSTM framework generally achieves the best forecasting performance in the dataset. Similarly, LSTM algorithm has also been applied to PV power prediction. As the main influence factor of PV power generation, the solar irradiance and its accurate forecasting are prerequisites for solar PV power forecasting. Authors in [24] proposed an improved LSTM model to enhance the accuracy of day-ahead solar irradiance forecasting, and the simulation results indicated that the proposed model has high superiority in the solar irradiance forecasting, especially under extreme weather conditions. A new method for 1-h-ahead PV power forecasting using deep LSTM networks was proposed [25], which can capture abstract concepts in the PV power sequences. The proposed method gave a very small forecasting error compared to the other methods. However, this paper did not incorporate environmental parameters, such as, wind speed, air temperature. Under certain extremely weather conditions, ever-increasing number of abnormal weather manifestations that are poorly predictable can be a complicating factor in the forecast of energy production. Therefore, various weather types and other meteorological parameters were considered in this paper.

Based on analysis above, a novel deep RNN models with LSTM units was proposed. In order to forecast the power of PV power output in 1-h-resolution over short-term time horizon, the model training process was treated as a sequence to sequence learning problem. Section 2 is based on methodology of the LSTM technique, whereas, Section 3 presents the implementation details, while related results are discussed in Section 4. The conclusion is drawn in Section 5.

## 2. Theoretical Background

### 2.1. Neural Networks

As one of the most important members in the field of machine learning [26], neural network is considered an effective prognostic algorithm. Multilayer perceptron, radial basis function networks, and other neural networks have been widely used in anomaly detection, damage clustering, and fault diagnosis. Excitingly, remarkable success has been achieved [27].

As for time series data, such as the samples in power forecasting problem, researchers have been searching for more reasonable models and algorithms to tap into their potential. Traditional static neural networks such as multi-layer perceptron (MLP) considered a sample point independently and create a prediction at each time step, and a majority of information in historical data was lost in the process. LSTM networks overcame the issue above and 'vanishing gradients' and 'exploding gradients' problems in RNNs.

### 2.2. Recurrent Neural Networks and Long Short-Term Memory

These transforms are inverted on forecasts to return them into their original scale before calculating and error score.

An RNN is defined as an ANN with arbitrary connections between neurons, usually fully connected between adjacent layers. It models temporal dependencies present in time series data

through the use of feedback connections and aims to 'remember' the values at previous time steps. As shown in Figure 2, it gives a simple RNN with one input unit, one output unit, and one recurrent hidden unit unfolded into a full network. $o_t$ is the output at time step $t$, $x_t$ is the input at time $t$, $s_t$ is the state at time step $t$, and it is the 'memory' of RNN. $W$, $U$, and $V$ are parameters in different network layers, and an RNN can share the same parameters ($W$, $U$, and $V$ above) across all steps [28]. The process can be expressed as

$$s_j(t) = f\left(\sum_i^l x_i(t)v_{ji} + \sum_h^m s_h(t-1)u_{jh} + b_j\right) \qquad (1)$$

where, $f(\cdot)$ is map function which is usually nonlinear such as tanh or rectified linear unit (ReLU). $\sum_i^l x_i(t)v_{ji}$ is the input layer at the time of $t$. $\sum_h^m s_h(t-1)u_{jh} + b_j$ is the input of hidden layer of time $(t-1)$. $b_j$ is the bias.

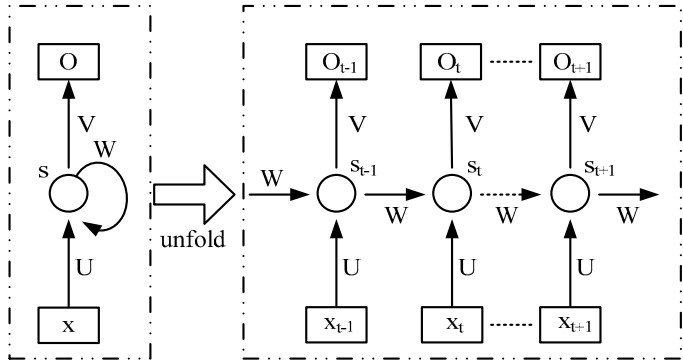

**Figure 2.** General overview of recurrent neural networks.

The LSTM algorithm belongs to the family of recurrent neural networks (RNN). RNNs commonly face a problem called the "exploding/vanishing gradients problem". LSTM is designed to overcome above question by creating a special type of structures called memory cells and gate units. So it is good for remembering information for long time.

To understand the functioning of LSTM memory units, Figure 3 depicts a single localized LSTM cell in the first layer of a network at a time step $t$. Module has the output values input ($i_t$), update ($g_t$), output ($o_t$) and forget ($f_t$) of four types of gates as shown in Figure 3.

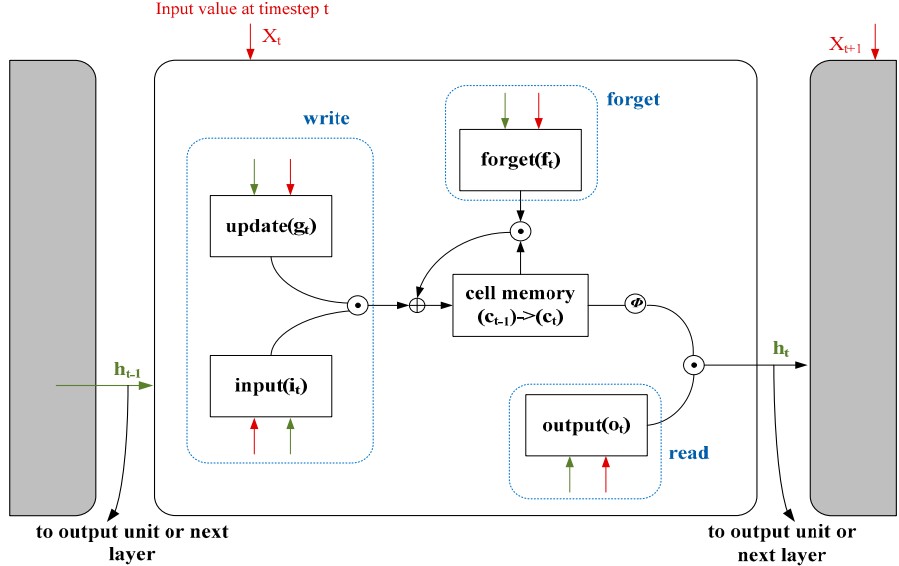

**Figure 3.** A LSTM (long short-term memory) unit in the first layer for time step (t).

First of all, the decision which information will be thrown away from the cell state is made by forget gate. Equations (2)–(6) are governing equations, which we explain below. Please note that $W_f$ and $U_f$ are weight matrices of input and output data from the previous LSTM cells. $b_n$ represents bias vector parameters which need to be learned during training. $f_t$ is the activation vector of forget gate. The sigmoid activation (linearly approximated standard sigmoid) has a range of 0 (forget all) to 1 (remember all) and is governed by Equation (2) [28].

$$f_t = sigm(W_f x_t + U_f h_{t-1} + b_f) \tag{2}$$

Next, once an input $x_t$ enters the LSTM cell, it is separately passed through an activation function, $g_t$ and input gate $i_t$. The cell can decide which values to write through the input gate and create a vector of new cell values through the update gate. Equations (3) and (4) represent their working, note $g_t$ and $i_t$ separately use hyperbolic tangent (tanh) and hard sigmoid (tanh) activation [28].

$$g_t = tanh(W_f x_t + U_f h_{t-1} + b_f) \tag{3}$$

$$i_t = sigm(W_i x_t + U_i h_{t-1} + b_i) \tag{4}$$

Cell memory (see Figure 2) updates itself recursively by interaction of its old value $(t-1)$ with forget and write gates' values. After these, combine these two parts to create an update to the states, "⊙" represents scalar multiplication. It follows (5) [28]

$$C_t = f_t \odot C_{t-1} + i_t \odot C_t \tag{5}$$

Lastly, the output gate decides which part of the cell states will be outputted by Equation (6). Ultimately, the output $h_t$ of LSTM unit at time $t$ can be obtained by applying Rectified Linear Unit (ReLU) to cell states through and multiplying it by output gate values as Equation (7) shows. Where $(W_0, b_0)$ are the weights and bias of output gate. $\phi$ is activation function (ReLU) [28].

$$o_t = sigm(W_0 x_t + U_0 h_{t-1} + b_0) \tag{6}$$

$$h_t = o_t \odot \phi(c_t) \tag{7}$$

According to the solar photovoltaic forecast LSTM structure above, the gradient of deviation and weight can be calculated everywhere. After that, put them into optimization algorithms Adam, the optimal value of cost function and the optimal solution can be available. In order to avoid arriving a local optimal solution, the training process is repeated for many times and the test set is separated from the training set for each trail.

### 2.3. Gradient Descent Algorithm

The ADAM algorithm is used to optimize the weights in each layer which exhibits faster convergence than the conventional stochastic gradient descent [29]. ADAM is a first-order based gradient descent optimization algorithm that is computationally efficient, and is suitable for optimizing models with a large set of parameters. Rather than naively updating the weights with a constant learning rate (as was the case for the vanilla stochastic gradient descent), ADAM considers the bias-corrected estimates of the moving average of the gradient as well as the squared gradient. Details on the ADAM algorithm can be found in other literature [30].

## 3. Model Description

### 3.1. Model Structure

The deep RNN models with LSTM units presented in this paper were developed in order to predict PV power values in 1-h resolution over 3 consecutive days. The proposed models were tested

in different seasons. A schematic block of the LSTM used to forecast the profile of the power produced by the PV plant is depicted. Figure 4 shows that there are four models for forecasting PV power in different seasons based on next 1 h of meteorological data.

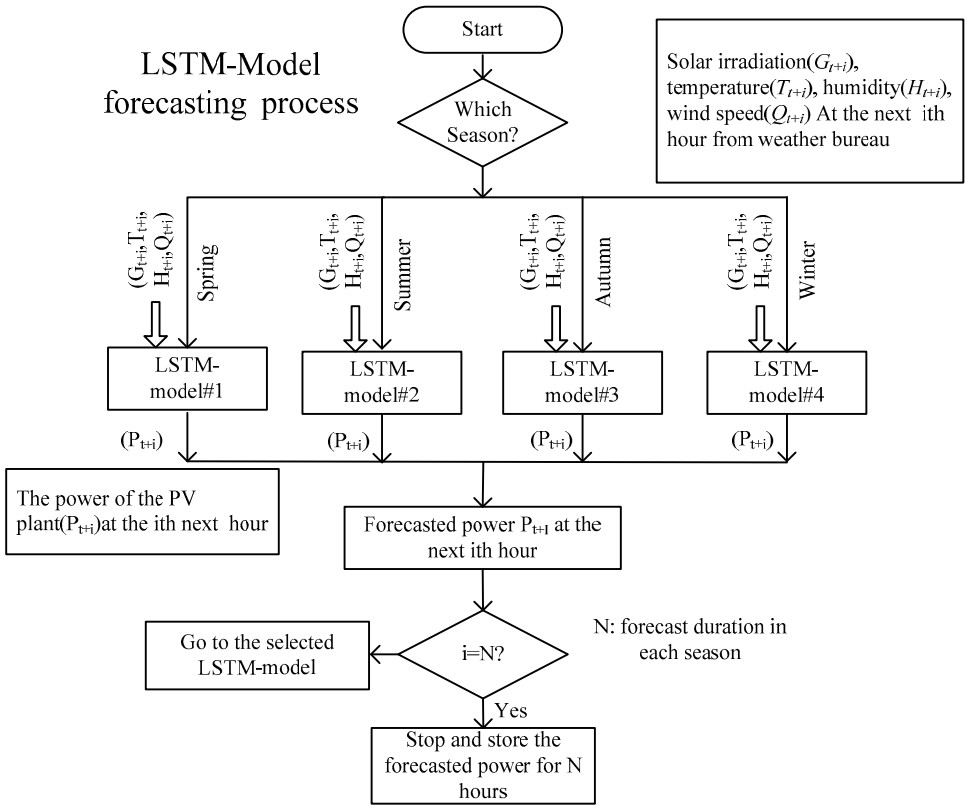

**Figure 4.** Flowchart of PV power forecasting design.

As for the structure of LSTM in Figure 5, the size of the input layer, the number of hidden layers, and the number of hidden units in each hidden layer are firstly determined. The detailed model is described as follows:

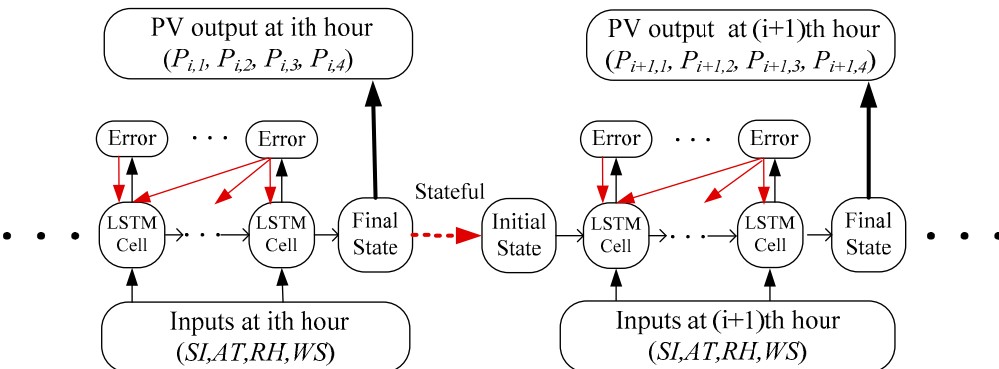

**Figure 5.** Diagram of PV power forecasting using LSTM.

For the input layer LSTM input: 3D tensor with shape (batch size, timesteps, input dim). The value of timesteps is 4, which can be determined by the set of data during 1 hour, and the input dim is 4 (solar irradiation (*SI*), air temperature (*AT*), relative humidity (*RH*), and wind speed (*WS*)).

For the hidden layer: The setting of the hidden layer is usually referred to the complexity of power generation process, and the model parameters are different for each season. The operation of LSTM is based on the batch. The 'stateful' in Figure 5 means there are state-related between batches. In the

LSTM model, sometimes a dropout is set to avoid over-fitting [31]. When a dropout is increased to more than 0.2, the predicted result is usually random and volatile.

For the LSTM output: final state at $i$th hour consists of two operations. In the first operation, the forecasting results in final state are passed to the initial state at next $(i + 1)$th hour. The short-term memory of previous results was achieved during this operation. In other words, the forecasting process in the last hour could be regarded as the training process at $(i + 1)$th hour. In the second operation, the results are passed to the output layer as the PV output $(P_{i,1}, P_{i,2}, P_{i,3}, P_{i,4})$ at $i$th hour. It requires attention that $P_{i+1,1}$ is the next sequential value of $P_{i,4}$. The prediction results for three consecutive days consist of hourly forecasting values.

### 3.2. Data Preparation

For the plant discussed in our work, the radiation angle and location of PV are fixed. Therefore, this information is blended in the historical power output data, with higher self-correlation than the indirect forecasting method. In this paper, the solar power data used in this study has been collected from a 10 MW peak solar power plant located at Jinan, China. The coordinates of this PV power plant in Jinan are 36 degrees 40 s north latitude and 117 degrees east longitude. The inputs of the proposed model were based on numerical weather predictions (NWPs), which were received from the meteorological services of Jinan. The NWPs include the solar irradiance, lowest air temperature, highest air temperature, relative humidity, wind speed, wind direction, cloud amounts, air pressure, and so on. The plant is facing south, the tilt angle is 25°, and the shading angle due to the presence of parallel rows of PV modules is 20°. The 10-MW plant utilizes a centralized inverter connected through a power transformer to the medium voltage grid. The strings are made of 20 series-connected PV modules, while groups of 16 strings are parallel-connected into 16 DC boards where fuses prevent over currents into the strings. The AC sides of the two inverters are connected to a double primary transformer that converts the low voltage output of the inverters (220 V) up to 500 kV, corresponding to the nominal voltage of the electrical grid. The test time period used for prediction is from 1 January 2017 to 31 December 2017. The data interval is 15 min, which is the requirement of grid in China. Figure 6 shows the solar radiation plays a significant role in PV power output forecasting. Relative humidity most strongly affects the production of electricity after the solar radiation parameter. As shown in Figure 7, the daily radiation time depends on different seasons. Therefore, the power of PV power generation is predicted in different seasons. Figure 8 shows the power output data in PV system in different days under four weather types: cloudy day, sunny day, foggy day, and rainy day, and it is obvious that the power output varies greatly under different weather conditions. Therefore, it puts forward higher requirements on the accuracy of LSTM.

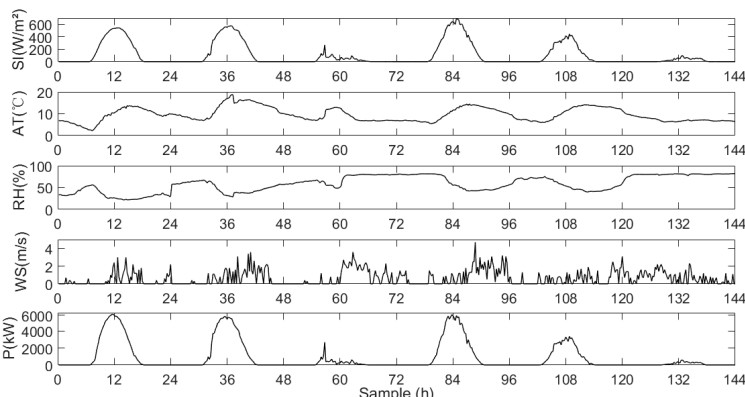

**Figure 6.** Solar irradiation (SI), air temperature (AT), relative humidity (RH), wind speed (WS), and PV power (P).

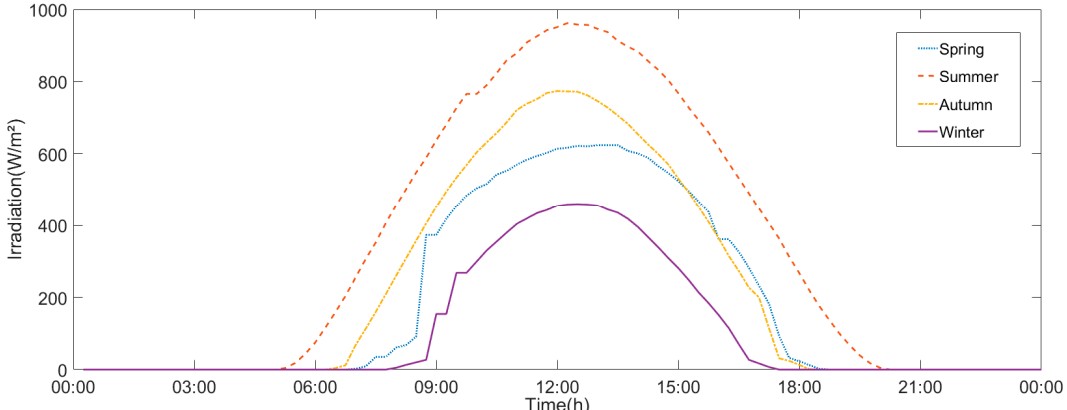

**Figure 7.** The four seasons' periods over diurnal variations in irradiance (spring: 12.3 h; summer: 14.7 h; autumn: 12.7 h; winter: 10.2 h).

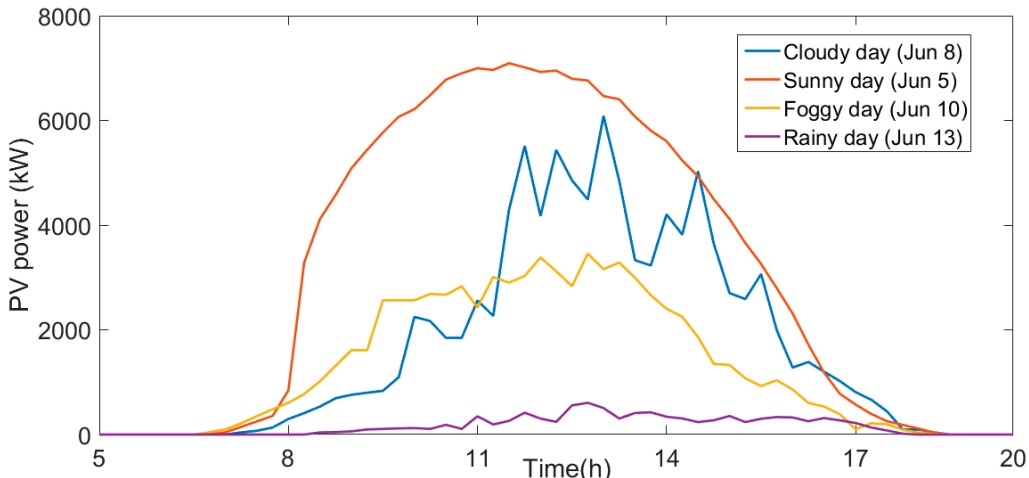

**Figure 8.** Photovoltaic system power output under different weather conditions.

In order to ensure the accuracy and speed of LSTM-models, it is necessary for data pre-processing. One of the well-known solutions is normalization on the basic of eliminating the anomalous data, by which the data can be restricted within the range between 0 and 1 to minimize the regression error, improve precision, and maintain correlation among data set. The process formula is shown as Equation (8)

$$Y_n = \frac{Y_n - Y_{\min}}{Y_{\max} - Y_{\min}} \tag{8}$$

where $Y_n$ is the original input data; $Y_{\min}$ and $Y_{\max}$ are the minimum and maximum input data.

### 3.3. Performance Criteria

In order to examine the capability of the designed LSTM-models to forecast the power produced by the PV plant, four seasons for each class (spring, summer, autumn, and winter) have been considered. Mean absolute percentage error (MAPE) and root mean square error (RMSE) as shown below will be used to evaluate the accuracy of forecasting and training process [1,5].

$$MAPE = \frac{1}{N}\sum_{t=1}^{N}\left|\frac{S_i - Y_i}{Y_{total}}\right| \times 100\% \tag{9}$$

$$RMSE = \frac{\sqrt{\frac{1}{n}\sum\limits_{t=1}^{N}(S_i - Y_i)^2}}{Y_m} \times 100\% \tag{10}$$

where, $S_i$ is the forecasting result at the time point $i$; $Y_i$ is the actual value at the time point $i$. $Y_{total}$ is the PV installation capacity; $Y_m$ is the mean power generation of each season. For the case in this paper, the value of $Y_{total}$ is 10 MW.

## 4. Evaluation and Discussions

### 4.1. Optimal Model Parameter Selection

As mentioned above, this study proposes 1-h-ahead forecasting of the power generation using long LSTM networks and compares with other algorithms. The summer power generation forecasting is taken as discussing example model, the training set (3136 exemplars from 1 June 2017 to 31 July 2017) is used to find the optimal parameters of the LSTM units, the validation set (147 exemplars from 1 August 2017 to 3 August 2017) is used to find the optimal layers and parameters, the test set ((147 exemplars from 4 August 2017 to 6 August 2017) is used to estimate RMSE and MAPE value after the final model has been determined. It should be pointed out that the power which is generated at night is ignored. As shown in Table 1, the detail dates for LSTM forecasting models are presented.

**Table 1.** Dates for LSTM (long-short term memory) forecasting models in all seasons.

| Dates for All Seasons | Training Set | Validation Set | Test Set |
|---|---|---|---|
| Spring | 1 March 2017 to 10 May 2017 | 11 May to 13 May 2017 | 14 May to 16 May 2017 |
| Summer | 1 June to 31 July 2017 | 1 August to 3 August 2017 | 4 August to 6 August 2017 |
| Autumn | 1 September to 31 October 2017 | 1 November to 3 November 2017 | 4 November to 6 November 2017 |
| Winter | 1 December 2017 to 5 February 2018 | 6 February to 8 February 2018 | 9 February to 11 February 2018 |

In order to determine the optimal model for photovoltaic generation forecasting, layers and parameters are adjustable to analyze the impact on forecasting accuracy. As shown in Table 2, RMSE and MAPE are calculated to evaluate the forecasting performance using validation set. The initial LSTM model we used has one input layer, two hidden layers (1 LSTM layer consisted of 30 neurons and 1 dense layer with 80 neurons), 1 output layer that makes a single value prediction. The default sigmoid activation function is used for the LSTM blocks and ADAM is chosen for the model optimizer. After adding to 2 LSTM layers with 10 neurons and 2 dense layers respectively with 70 and 80 neurons, the RMSE value ranges from 11.26% to 8.21%. The forecasting performance has been improved. When the number of LSTM layers increases to 3, RMSE value increases from 8.21% to 12.92%. As the number of layers increases, the training model will have a better effect, even reaching 100% prediction accuracy. However, what follows is that the model is over-fitted, and the prediction effect is seriously reduced when the model is put on the test data. Above all, increasing the number of hidden layers may be not beneficial to forecasting performance. The hidden layer structure that has 2 LSTM layers with 10 neurons and 2 dense layers respectively with 70 and 80 neurons are determined as the final model for summer. The optimal structures for other seasons are shown in Table 3.

**Table 2.** Comparison of different model structures for summer in RMSE (root mean square error) and MAPE (mean absolute percentage error).

| Layers and Parameters | | | |
|---|---|---|---|
| **Layers and Neurons of LSTM** | **Layers and Neurons of Dense** | **RMSE** | **MAPE** |
| 1 (30) | 1 (80) | 11.26% | 2.72% |
| 2 (10, 10) | 2 (70, 80) | 8.21% | 2.01% |
| 3 (10, 10, 10) | 1 (80) | 11.18% | 2.76% |
| 4 (10, 10, 10, 30) | 2 (80, 80) | 12.30% | 2.95% |

**Table 3.** Parameters used for the LSTM in each season.

| LSTM-Model | Learning Rate | Optimizer | Layers and Neurons of LSTM | Layers and Neurons of Dense | Dropout |
|---|---|---|---|---|---|
| #1 (Spring) | 0.01 | ADAM | 3 (10, 10, 10) | 1 (80) | 0.1 |
| #2 (Summer) | 0.01 | ADAM | 2 (10, 10) | 2 (70, 80) | \ |
| #3 (Autumn) | 0.15 | ADAM | 2 (10, 10) | 3 (70, 80, 80) | \ |
| #4 (Winter) | 0.01 | ADAM | 3 (10, 20, 20) | 1 (80) | \ |

## 4.2. Forecasting Result and Discussion

The training trends of each algorithm are shown in Figure 9 and the error analysis of each algorithm are shown in Table 4, the RMSE and MAPE values for LSTM are respectively 4.07% and 1.52%, while the RMSE and MAPE values for back propagation (BP) networks, least squares support vector machine (LSSVM) and wavelet neural (WN) networks are respectively 16.61% and 2.82%, 16.38% and 3.44%, and 21.29% and 4.83%. The obvious low values indicate that the training result of the LSTM model has a high level of accuracy.

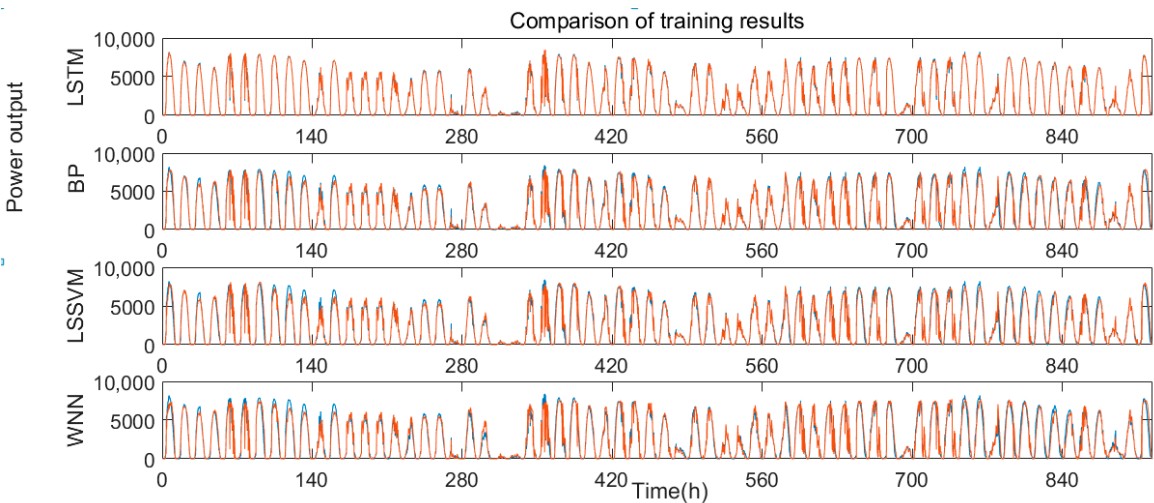

**Figure 9.** Comparison of training results of different algorithms and actual values in summer (from 1 June to 31 July 2017).

**Table 4.** RMSE and MAPE of training and forecasting in every season.

| Season | | BP RMSE (%) | BP MAPE (%) | LSSVM RMSE (%) | LSSVM MAPE (%) | WNN RMSE (%) | WNN MAPE (%) | LSTM RMSE (%) | LSTM MAPE (%) |
|---|---|---|---|---|---|---|---|---|---|
| Spring | training | 17.5% | 2.86% | 18.6% | 3.08% | 21.02% | 3.69% | 5.09% | 0.82% |
| | forecasting | 19.6% | 4.82% | 20.1% | 5.37% | 18.8% | 4.8% | 5.34% | 1.51% |
| Summer | training | 16.61% | 2.82% | 16.38% | 3.44% | 21.29% | 4.83% | 4.07% | 1.52% |
| | forecasting | 13.03% | 3.18% | 13.3% | 2.85% | 19.05% | 4.68% | 9.57% | 2.01% |
| Autumn | training | 21.91% | 3.14% | 21.71% | 3.28% | 26.99% | 4.38% | 4.96% | 1.06% |
| | forecasting | 20.94% | 2.43% | 23.11% | 2.37% | 23.68% | 2.79% | 13.86% | 1.51% |
| Winter | training | 24.05% | 2.45% | 24.68% | 2.42% | 29.85% | 3.16% | 3.38% | 0.38% |
| | forecasting | 24.68% | 5.47% | 17.74% | 3.69% | 25.45% | 6% | 9.26% | 1.38% |

Different model structures have been analyzed and the optimal model for each season has been determined in the above. Figure 10b,c are the partial enlarged views of Figure 10a. In Figure 10b, it can be observed that the fitting performance using LSTM is better than using other algorithms with a long

delay in the first six hours. There are some unknown down trend which cannot accurately forecast actual power trends. As shown in Figure 10c, in the case of large weather fluctuations, the power could be effectively predicted by all proposed algorithms, but the forecasting results using LSTM have little error compared to actual power because of the ability of faster perceiving changes from historical data, while the values of prediction using other algorithms are much smaller than the actual value. It can be concluded that the forecasting method using LSTM shows better performance with no delay in sunny days and less error in cloudy day. The same prediction situations appear in the following three figures.

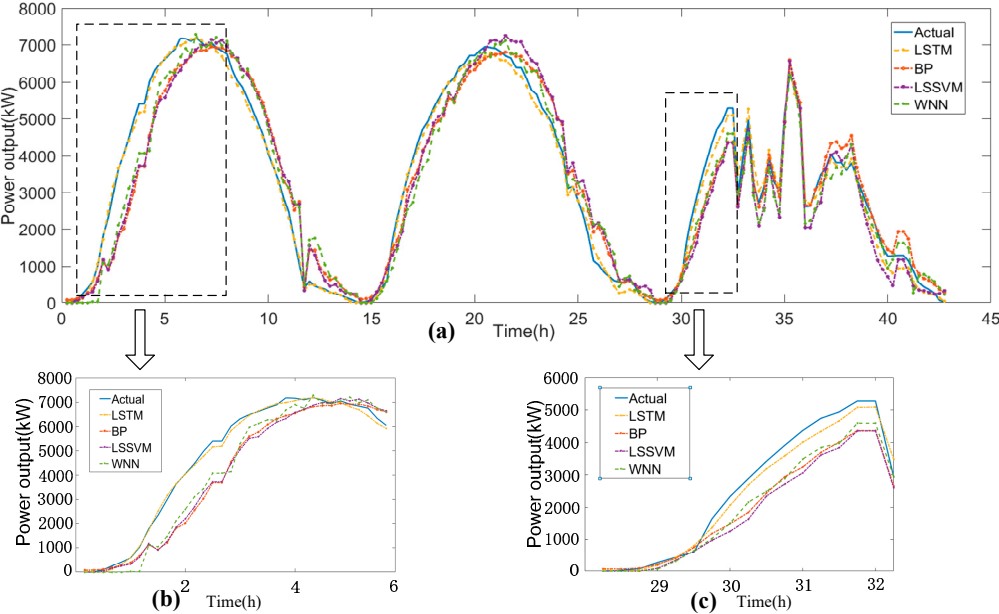

**Figure 10.** Short-term PV power forecasting in spring by various algorithms. (**a**): the general forecasting power trend of three days (**b**): the trend in the first 5 hours in (a) (**c**): the trend from 28th to 32th hour in (a).

Figures 10a, 11, 12 and 13 show forecasting performances for three consecutive days using optimal LSTM model in each season and comparisons with BP networks, LSSVM and WNN networks. As shown in Figure 12. In the periods of 2nd to 5th hour and the 25th to 28th hour, except LSTM algorithm, the predicted trends of other algorithms showed larger error with the actual power trend. Its RMSE value could reach 13.86%, which is higher than other seasons. In Figure 13 starting from the 13th hour, the results predicted by other algorithms lag behind the actual ones. After the 30th hour, the same error appears in the LSTM model. It can be seen that the prediction accuracy under ideal weather type is not high, because the diversity of power generation trends under this type is lacked in training set. It can be observed that LSTM model exhibits optimal forecasting in each season and shows the similarly good forecasting performance for non-ideal weather types compared to the ideal weather type. RMSE and MAPE values of forecasting are shown in Table 4. It shows that the RMSE values of forecasting using LSTM models in four seasons are respectively 5.34%, 9.57%, 13.86%, and 9.26%, while the lowest values using other algorithms are respectively 18.8%, 13.03%, 20.94%, and 17.74%. In addition, it demonstrates that the forecasting results in autumn are not particularly ideal with RMSE value of 13.86%. However, the LSTM model still shows the best performance compared with other algorithms.

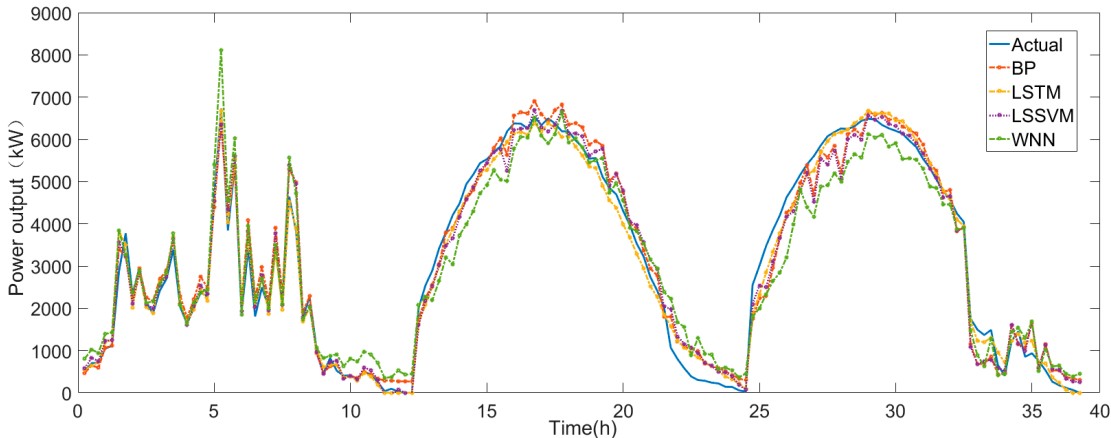

**Figure 11.** Short-term PV power forecasting in summer by various algorithms.

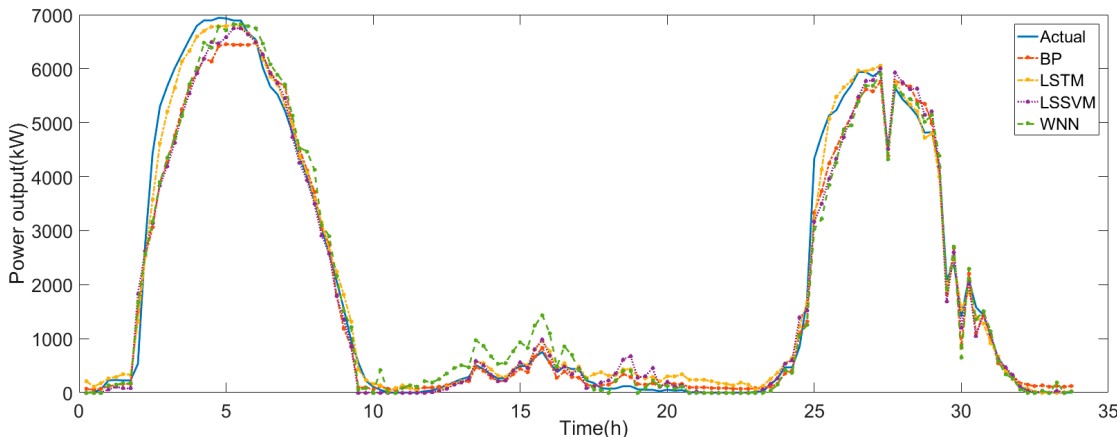

**Figure 12.** Short-term PV power forecasting in autumn by various algorithms.

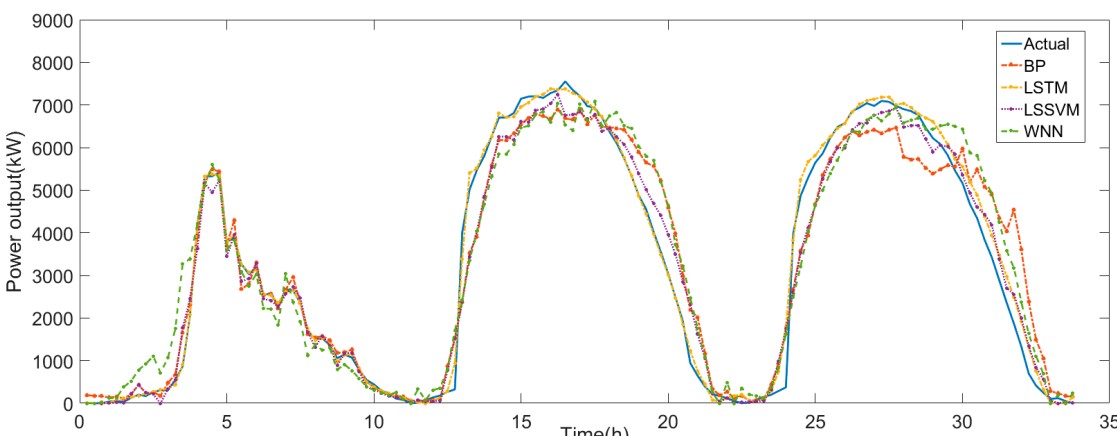

**Figure 13.** Short-term PV power forecasting in winter by various algorithms.

## 5. Conclusions

This paper discussed the LSTM algorithm theoretically and the method was applied in PV power output time series forecasting. Without changing the input layer, optimizer, output layer, and learning rate, this paper discusses the influence on forecasting performance by changing parameters and the number of hidden layers. The developed model is versatile, and that can be used to forecast the solar PV power output without any complicated calculations for any region provided comprehensive historical power and meteorological data. The final LSTM model for four seasons is established by a

training set and evaluated by a validation set. Finally, 1-h-ahead forecasting of the power generation is achieved by a test set.

Furthermore, this paper describes and compares four short-term forecasting models (LSTM, BP networks, WN networks, and LSSVM) for 1-h-ahead forecasting of the power generation using meteorological data in next hour. The RMSE values of forecasting using LSTM models in four seasons are 5.34%, 9.57%, 13.86%, and 9.26%, respectively. While the lowest values using other algorithms are respectively 18.8%, 13.03%, 20.94%, and 17.74%, respectively. The result shows that the accuracy using the final LSTM model for forecasting is sufficiently high and that it can be used in instantaneous control of micro grids. A PV power producer is economically more penalized when larger derivation (error) exists between real power productions and electric energy sale bids of the PV plant. Therefore, the method proposed in this paper contributes to the reduction of economic penalizations in PV plant owners' retributions and, therefore, to increasing net profits for PV plant owners.

**Author Contributions:** Conceptualization, M.G. and F.H.; M.G. and F.H.; software, J.L. and F.H.; validation, J.L, formal analysis, J.L. and F.H.; investigation, F.H. and J.L.; resources, F.H.; data curation, J.L.; writing—original draft preparation, J.L. and F.H.; writing—review and editing, J.L., F.H. and D.L.; visualization, J.L.; supervision, J.L. and F.H.; project administration, M.G.; funding acquisition, M.G.

**Funding:** This project was funded by the Project Supported by the Fundamental Research Funds for the Central Universities (2018ZD05).

**Conflicts of Interest:** The authors declare no conflict of interest.

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
