# Peer review of "Short-Term Forecasting of Power Production in a Large-Scale Photovoltaic Plant Based on LSTM"

_applsci, doi:10.3390/app9153192_

Round 1

Reviewer 1 Report

General Comments:

The application of the combination of recurrent neural networks with a long short-term memory model is novel to PV short-term forecasting.

Main comments

(i) On page 16 in Section 4.1 Optimal model parameter selection, dates of the data for training the models for the summer period were indicated (line 296). Also include, the dates for the data used in training the models for all seasons.

(ii) On page 17, the authors state that forecasting accuracy does not benefit from increasing the number of hidden layers. Does the accuracy decrease (or plateau) instead? If so, explicitly state that the accuracy decreases (or plateaus). Also, offer a reason for this.

(iii) On page 21, Table 3: The error metric RMSE is higher for the LSTM in autumn and winter. First, state the result in the text and then offer an explanation.

(iv) Are there any specific situations that the models will not apply? Please discuss, if any, the limitations of the approach taken.

(v) In the conclusion, page 22, lines 382-383: The authors state that the method that they propose assists in reducing the economic penalties that a PV plant owner will face. Explain this further in terms of how such reductions could benefit the wider public.

Comments

(vi) On page 4, line 73, explain what MAPE stands for.

(vii) The paragraph on lines 123 - 125 should be joined with the previous paragraph.

(viii) Include relevant citations for each equation on pages 7,  9, 15 and 16.

(ix) What do batch_size and input_dim refer to on lines 223 and 224, respectively?

(x) In the data section (page 13), include the coordinates of the plant and where the meteorological conditions were monitored.

(xi) Please include the dates of the data observations in the caption of Fig. 6. Also, include the period over which the diurnal variations in irradiance are computed in the caption of Figure 7.

(xii) On line 262, include sample dates for each of the four weather conditions for which data was used to create Fig. 8.

(xiii) A number of error metrics exist. Why were only the MAPE and RMSE used to assess the performance of the models?

(xiv) On page 17, Line 315-316: The analysis is in reference to the model for the summer period. This seems to be in reference to Table 1. Therefore indicate the season in Table 1's caption.

(xv) On page 18: Figure 9: Indicate the data dates in the caption of the figure.

(xvi) Page 21, Lines 363-364: The sentence is not clear. Please revise it.

Author Response

   We are grateful for your detailed comments on the revision. Details are modified as shown in the following documents

Reviewer 2 Report

The relevance of the issues concerned does not raise questions. Alternative energy is increasingly used in both developed and developing countries. Therefore, methods for predicting favorable conditions are extremely important. Climate change and the ever-increasing number of abnormal weather manifestations that are poorly predictable can be a complicating factor in the forecast of energy production due to the sun. The forecast of summer convective clouds can also significantly distort and worsen the forecast. About this little said in the article. On the other hand, figures 10-12 show that there is no big difference in the forecast of electricity generation between the existing and the proposed models. Variations in the accuracy of the forecast of cloudiness and precipitation are much higher than difference between accuracy of existed and proposed models for forecast of PV power. At the same time, precipitation and cloudiness most strongly affect the production of electricity after the solar radiation parameter. Above needs some clarification.

Author Response

We are very grateful for your comments. As you said, precipitation and cloudiness are both important factors affecting power generation, especially in summer. In this study, the influence of precipitation and cloudiness on the forecasting of power generation is characterized by solar irradiation and relative humidity respectively. The input of the proposed method is based on the feedback of weather information for the next hour from NWPs. And the meteorological data is updated every 15 minutes to ensure the accuracy of its input variables. In most other related papers, the prediction duration has a great impact on the prediction accuracy of power trend.Therefore, in 1-h ahead PV power prediction, each algorithm can forecast the trend of general power generation. Moreover, the forecasting results using LSTM model performs better than others, because it has memory function for the previous time series power outputs.

Reviewer 3 Report

After a careful reading I cannot understand what is the innovation brought in this paper.

Furthermore the paper should be better structured, by eliminating the inessential parts and focusing the discussion on the developed parts of the authors.

Author Response

Thank you very much for your attention and the referee’s evaluation and comments on our paper. Our paper surely has some shortcomings and we hope that the revised version can meet the level requirements of publication. Inessential parts have been eliminated and the developed parts have been focused in the revised paper.

Reviewer 4 Report

The paper is well written and easy to follow. The research is sound and follows an appropiate methodology.

Authors need to add a related work section to their paper and discuss similar works, comparing their approach to others. Authors should discuss two main areas in that section, other approaches for power production forecasting and the usage of LSTMs to forecasting in other domains. Some papers that authors can use in the section:

Gensler, A., Henze, J., Sick, B., & Raabe, N. (2016, October). Deep Learning for solar power forecasting—An approach using AutoEncoder and LSTM Neural Networks. In 2016 IEEE international conference on systems, man, and cybernetics (SMC) (pp. 002858-002865). IEEE.

Abdel-Nasser, M., & Mahmoud, K. (2017). Accurate photovoltaic power forecasting models using deep LSTM-RNN. Neural Computing and Applications, 1-14.

Kong, W., Dong, Z. Y., Jia, Y., Hill, D. J., Xu, Y., & Zhang, Y. (2017). Short-term residential load forecasting based on LSTM recurrent neural network. IEEE Transactions on Smart Grid, 10(1), 841-851.

Almeida, A., & Azkune, G. (2018). Predicting human behaviour with recurrent neural networks. Applied Sciences, 8(2), 305.

Wang, F., Yu, Y., Zhang, Z., Li, J., Zhen, Z., & Li, K. (2018). Wavelet decomposition and convolutional LSTM networks based improved deep learning model for solar irradiance forecasting. Applied Sciences, 8(8), 1286.

Kavitha, S., Mohanavalli, S., & Bharathi, B. (2018, November). Predicting Learning Behaviour of Online Course Learners' using Hybrid Deep Learning Model. In 2018 IEEE 6th International Conference on MOOCs, Innovation and Technology in Education (MITE) (pp. 98-102). IEEE.

It is important that in this section the authors make clear their contributions to the state of the art.

Author Response

   Thank you very much for your suggestion. This paper of previous version lacked some introduction about the research and application about the LSTM algorithm. I have read all papers you provided and summarize the works of these papers. Some content of this related works has been added to the section of Introduction, the content is shown as follows.

   Nowadays, LSTM algorithm has been applied to various fields, including human behavior predicting, short-term residential load forecasting and renewable energy. Aiming at early detection of the risks related to mild cognitive impairment and frailty and providing meaningful interventions that prevent these risks, [22] have created a deep learning architecture based on LSTM to predict the user’s next actions and to identify anomalous user behaviors. It was also applied in residential load forecasting. The LSTM-based framework was proposed to address the short-term load forecasting problem including high volatility and uncertainty for individual residential households[23]. The proposed LSTM framework achieves generally the best forecasting performance in the dataset. Similarly, LSTM algorithm has also been applied to PV power prediction. As the main influence factor of PV power generation, the solar irradiance and its accurate forecasting are prerequisites for solar PV power forecasting. [24] proposed an improved LSTM model to enhance the accuracy of day-ahead solar irradiance forecasting, and the simulation results indicated that the proposed model has high superiority in the solar irradiance forecasting, especially under extreme weather conditions. A new method for 1h-ahead PV power forecasting using deep LSTM networks was proposed [25], which can capture abstract concepts in the PV power sequences. The proposed method gave a very small forecasting error compared to the other methods. But this paper did not incorporate environmental parameters, such as, wind speed, air temperature. Under certain extremely weather conditions, ever-increasing number of abnormal weather manifestations that are poorly predictable can be a complicating factor in the forecast of energy production. So various weather types and other meteorological parameters were considered in this paper.

Round 2

Reviewer 3 Report

In this form the paper can be accepted.